# Canine Spermatozoa—Predictability of Cryotolerance

**DOI:** 10.3390/ani12060733

**Published:** 2022-03-15

**Authors:** Sabine Schäfer-Somi, Martina Colombo, Gaia Cecilia Luvoni

**Affiliations:** 1Department for Small Animals and Horses, Platform for Artificial Insemination and Embryo Transfer, University of Veterinary Medicine, 1210 Vienna, Austria; 2Dipartimento di Medicina Veterinaria e Scienze Animali (DIVAS), Università degli Studi di Milano, 26900 Lodi, Italy; martina.colombo@unimi.it (M.C.); cecilia.luvoni@unimi.it (G.C.L.)

**Keywords:** dogs, semen, cryopreservation, markers

## Abstract

**Simple Summary:**

Semen freezing in dogs is a field of growing interest. The international shipment of cryoconserved semen contributes to the avoidance of long travels and long-term storage of valuable gametes. However, the collection of one semen portion on average results in one to three doses for artificial insemination, which is a poor result in comparison to the outcome in large animals. The costs for the owners are therefore rather high. In individual dogs, the semen quality of raw semen is good; however, it could be suboptimal after thawing. To avoid costly freezing of these low-quality ejaculates, markers of freezability are useful. An abundance of markers are available for large animals, but not for dogs. This review provides an overview on markers for freezability of canine semen.

**Abstract:**

Markers of freezability allow the selection of ejaculates of good freezability. So far, most investigations were conducted in boars, bulls, rams and horses, with high economic interests triggering the efforts. The progress in dogs is comparably slow. A critical evaluation of the methods requires consideration of practicability, with most labs not even possessing a computer assisted sperm analyser (CASA); furthermore, small canine ejaculates mostly do not allow the use of large semen volumes. In dogs, modern markers of freezability no longer assess single membrane constituents or seminal plasma components but comprise tests of cell functionality and adaptability, energy metabolism, cluster analyses of kinetic and morphometric parameters, as well as DNA intactness. Identification of the most efficient combination of tests seems useful. At present, examination by CASA combined with cluster analysis of kinetic subgroups, JC-1 staining and COMET assay or staining with toluidine blue seem most appropriate; however, cell volumetry and other functional tests deserve better attention. A better understanding of spermatozoa energy metabolism might reveal new markers. This review focuses on the requirements and markers of freezability of canine semen, highlighting potential future candidates.

## 1. Introduction

The freezability of a canine spermatozoa is determined by the complex interplay of sperm membrane and seminal plasma, diluent and cooling-freezing-thawing protocols [1,2,3,4,5,6], and it is influenced by further parameters such as age [7]. The sensitivity to cryoinjuries is species-specific, and recent reviews comprehensively summarized the requirements for successful cryopreservation of canine spermatozoa [1,8]. Canine spermatozoa membranes are less sensitive to cold injury than, for example, boar spermatozoa, due to a relatively high cholesterol:phospholipid ratio [9] and the polyunsaturated fatty acid (PUFA) content of the membrane. This is, among others, indicated by the fact that DNA integrity is not affected by freezing thawing [10], whereas in stallions, it is [11]. Unfortunately, the exact composition of the sperm membrane PUFA has not yet been unraveled in dogs, as in other species [12]. However, semen quality after thawing is, in addition, individual dependent. In some dogs, despite good quality of raw semen, post-thaw semen quality is bad [13,14,15,16]. This rectifies the intense search for prognostic markers of freezability. To date, mainly kinematic and morphometric parameters including cluster analyses [17,18,19], functional assays such as volumetry [20], the hypoosmotic swelling test (HOST, [21]) or the reaction to (Ca^2+^) ionophore treatment [22] and seminal plasma components [16,23] have been investigated; however, the usefulness for prediction of post-thaw semen quality has not been reviewed in a comparative manner so far. In dogs, the best results were achieved with functional tests, such as kinematic assays combined with cluster analyses, cell volumetry and the reaction to (Ca^2+^) ionophore, whereas the HOST was not useful for predicting post-thaw semen quality [21]. This is comparable to other species [24] and highlights the need for complex assays, investigating more than adaptability to changing osmolarities.

Meanwhile, in other species, an abundance of markers for freezability have been detected that have not yet been investigated in canines. Particularly in boars, the investigations are intense (for review: [25]). Only recently, some spermatozoa components of metabolic pathways related to energy metabolism were identified as probable suitable markers [26,27]. Another study using transcriptome analysis revealed an upregulation of several genes, and finally a higher concentration of three proteins related to inflammation and apoptosis in bad freezer ejaculates [28]. Comparable studies in dogs are lacking. In bulls, proteomic approaches revealed higher concentration of Arylsufatase A in semen of good freezers, which is indicative of the cell’s energy metabolism. Further markers of sperm functionality were assessed in high-density spermatozoa, indicative of normal glycolysis, zona binding and motility [29]. Meanwhile, in this species, genome-wide association studies investigated genetic causes of poor sperm quality, and microsatellites markers (SNPs), such as BM1500 and UMN2008, were found to be related to freezability, especially post-thaw motility [30]. In rams, a strong correlation between seminal plasma composition and freezing resilience of semen has been found [31]. In stallions, among others, caspase 3 activity and the lipoperoxidative status of semen were found to be suitable for the prediction of freezability [32];, as well as the content of cysteine-rich secretory proteins (CRISP-3) participating in sperm maturation [33]. In men, spermatozoa out of bad freezer ejaculates contained less enolase 1 and glucose-6-phosphate isomerase (GPI) than cells from good freezer ejaculates. Both parameters are among others indicators of glycolysis and energy production [34]. Furthermore, viscosity of fresh semen was negatively related to post-thaw motility and acrosome integrity, and low citric acid concentration was negatively related to post-thaw acrosome integrity [35].

Markers of freezability are thus mainly indicators of spermatozoa functionality and energy metabolism or seminal plasma components. The present review discusses the usefulness and reliability of markers of freezability of canine spermatozoa and highlights potential factors of future interest.

## 2. Cryopreservation and Membrane Damages

Cryopreservation of canine spermatozoa requires high adaptability of the cell to changing osmolarity and temperature. As in other species, the fluidity of the membrane changes significantly during the freeze-thawing procedure, which coincides with a re-arrangement of membrane phospholipids. The membrane fluidity is dependent on the cholesterol content and the amount of disulfide bonds, the acyl chain length saturation and the temperature of the surrounding milieu. Very high contents of cholesterol and low contents of polyunsaturated fatty acids (PUFA) make the membrane more rigid, especially at low temperatures and, consequently, may result in leaky membrane [36]. Cholesterol helps to maintain membrane intactness at non-cryogenic temperatures, primarily during cold shock, which is usually at above-freezing temperatures [37]. When the temperature decreases, the membrane lipids change toward the crystalline phase with lateral segregation, lipid peroxidation, loss of lipids and formation of reactive oxygen species [38]. This finally leads to membrane destabilization and may cause membrane damage, especially when the cooling rates during freezing are too high or too low [39,40,41]. In canines, moderate freezing rates have been shown to be advantageous (−10 to −40/min, [42]). Cell damage occurs due to ice crystal formation, concentration of solutes, electrolytes and cell dehydration. Cryodamage, which is comparable among species, causes, among others, a severe loss of essential membrane proteins and receptors, degenerative acrosome exocytosis, degradation of mRNAs, disruption of the perinuclear theca, reduction in mitochondrial activity, changes in membrane fluidity/integrity and in ion channels, reduction in sperm motility, disruption of disulfide bridges between cysteine radicals of protamines and DNA fragmentation [25]. Increased cholesterol efflux and a loss of potassium further decrease the fertilizing ability of the cell by causing preterm capacitation and acrosome reaction [39,40].

## 3. What Makes Sperm Freezable?

Figure 1 provides an overview of the most important parameters and requirements, conserved among species.

### 3.1. The Composition of the Sperm Cell Membrane

The composition of the sperm cell membrane plays a fundamental role. Sperm membrane is a bilayer of mainly phospholipids, such as cholesterol and saturated, as well as poly-unsaturated, fatty acids. The composition shows broad species-specific differences, as well as individual differences. A relatively high cholesterol:polyunsaturated fatty acid ratio, such as in canines, was among others shown to be advantageous for cryotolerance [43,44]. We recently found a similar expression of cholesterol transport molecules before freezing (ATP-binding cassette transporter A1; ABCA1) in spermatozoa membranes from canine good and bad freezers. However, the bad freezers had lower seminal plasma concentrations of cholesterol [16]. The incorporation of cholesterol in the sperm membrane via cyclodextrins was shown to improve post-thaw semen quality in dogs [45], rams [46] and stallions [47,48], emphasizing the importance of the membrane cholesterol content. 

### 3.2. Membrane Intactness

Membrane intactness is the utmost prerequisite for normal post-thaw cell function. The interplay of seminal plasma components, such as cholesterol, peptides, hormones, membrane vesicles and enzymes, with the spermatozoa membrane is an important field of current investigation; however, there are some differences between species [49,50,51,52]. A loss of surface molecules, such as the progesterone receptor and proteins, especially from the acrosome, may impair post-thaw quality and fertilizing capacity of the cell [49]. An assessment of membrane integrity is therefore a useful part of pre-freeze semen evaluation. Fluorescent dys, such as SYBR-14/PI and CFDA enable visualization of damaged membranes [53], whereas an evaluation of specifically acrosome damages with fluoresceinated lectin peanut agglutinin was found to be extremely valuable to assess sperm quality before freezing and after thawing in dogs and other species [20,54].

Furthermore, the cell’s ability to maintain its energetic homeostasis, i.e., the ability of sufficient ATP formation in parallel to constant energy loss because of cell activity, is important. This is, among others, dependent on functional membrane hexose transporters. In dogs, the hexose transporters Glut 3 and 5 have been detected in intact spermatozoa membranes and were shown to increase the ATP formation very rapidly [55]; the localization and intensity of expression was modified by capacitation, which emphasizes the necessity of a proper functioning of the intact membrane [56].

The adaptability to changing osmotic conditions is of utmost importance, since during freezing, the cell has to release water and will shrink to avoid ice crystal formation, with consecutive concentration of solutes and electrolytes. A defective membrane will, among others, affect the function of principal water channel molecules. Aquaporins (AQP) exert a protective effect during the freeze-thaw procedure, not only by active osmoregulation but also by regulating the transport of small solutes, such as cryoprotective agents (CPA) [57]. They are widely distributed in testicular and epididymal tissue [58] and were detected in membranes of human, mouse, rat, boar, stallion, seabream, goose and bull spermatozoa (for review: [57]), although in a species-specific and sometimes even individual manner.

The impact of water channel molecules on semen freezability has been investigated in other species [59], and the expression of different AQP was related to the cells’ ability to survive the cryopreservation procedure, which seems to be species specific [57]. Recently, AQP-1 has been detected in the membrane of canine spermatozoa and was localized in the head, midpiece and tail [60]. In another study, AQP-8 was detected in canine spermatozoa by means of immunoblotting [61]. Unfortunately, an assessment of the site of expression was, thereby, not possible. The relative amount of AQP-8 was positively correlated with the percentage of proximal cytoplasmic droplets [61]. Being an orthodox AQP, AQP-8 is supposed to mainly contribute to cell volume regulation; however, it is not only permeable to water, and mitochondrial AQP-8 is, in addition, able to transport hydrogen peroxide. A role for AQP-8 in diffusion of ROS after cryopreservation has been suggested previously [57]. The function of AQP-8 in canine spermatozoa membranes and its usefulness as a marker of freezability remain to be investigated.

### 3.3. Energy Management

Another sensitive factor is the maintenance of mitochondria intactness, which is essential for a normal cell kinematic after freeze-thawing. In boar spermatozoa, these organelles are believed to be most sensitive to cryodamage [62]. In dogs, like in men, the inner mitochondrial membrane potential (IMM) of spermatozoa was found to correlate more strongly with membrane viability than with cell motility [63]; in some immotile sperm with high IMM, energy supply by oxidative phosphorylation is sufficient to survive but not sufficient to support motility—hexose metabolism [55] or gluconeogenesis from lactate and pyruvate [64] are part of a complex glycogen metabolism, additionally comprising energy storage in canine spermatozoa [64]. This may direct the interest toward markers of glycolysis, such as lactate, in dogs. However, the assessment of IMM by use of the JC-1 probe seems to be a good indicator of mitochondria function and oxidative ATP production before freezing. Further, mitochondria activity assays are commercially available but have mostly not been investigated as markers of freezability.

### 3.4. DNA Stability

Finally, DNA stability plays an important role. The stability of DNA after freeze thawing of spermatozoa shows species-specific differences, which was shown to be related to the cysteine residues in protamine 1 or an unbalanced protamine 1:2 ratio [65]. In dogs, we could show that freezing and thawing did not increase the rate of DNA fragmentation when evaluated immediately after thawing; however, the DNA fragmentation of thawed spermatozoa increased significantly within 3 h of storage at +37 °C [66]. Centrifugation before freezing increased the rate of DNA fragmentation immediately after thawing, which might explain the different findings of others [67]. Nevertheless, an assessment of DNA integrity before freezing is useful and can be performed by using the terminal deoxynucleotidyl transferase-mediated fluorescein-dUTP nick end labeling (TUNEL) test, the COMET assay [68] or the sperm chromatin structure assay (SCSA; [66,69,70]). Although flow cytometric measurements provide more statistical power because of higher cell numbers, the microscopic evaluation of single samples using COMET or TUNEL assay before freezing seems useful. Staining with toluidine blue was found to be fast and exact to assess the degree of DNA damage [71], also in canine spermatozoa [72]. This dye has great affinity to the free phosphate groups of DNA and protamines, which are increased in damaged DNA. However, a rather promising sophisticated approach is the Raman spectrometer. This technique relies on the interaction of light photons with molecules, resulting in different light-scattering patterns that can be immediately evaluated using semen or seminal fluid. Some applications include the differentiation of abnormal and normal seminal plasma and DNA damage [73].

### 3.5. The Composition of the Seminal Plasma

The seminal plasma plays a fundamental role in membrane function and adaptability; its composition shows manifold inter-species differences, and the effect of seminal plasma during freeze thawing is discussed controversially. In dogs, we recently found higher concentrations of cholesterol in seminal plasma of good freezers than in seminal plasma of bad freezers [16]. In another study, the removal of seminal plasma from ejaculates of good quality decreased post-thaw motility, increased the percentage of morphologically abnormal sperm and increased the DNA damage during 3 h of post-thaw storage [66]. Centrifugation evidently removes or decreases some ingredients otherwise contributing to the post-thaw condition of the membrane. A recent proteomic study revealed an abundance of proteins, including sperm membrane derived proteins, among others, related to cellular function, metabolism, maturation, binding, antioxidant capacity and intercellular action [74]. Seminal plasma contains manifold natural antioxidants, exerting a measurable total antioxidant capacity that can be highly variable and was found to be decreased in infertile dogs [23]. Post-thaw oxidative stress is one of the most detrimental parameters on spermatozoa membranes. Seminal plasma was found to exert a highly protective effect on cell mitochondria, especially hydrogen peroxide and hydroxyl radical; centrifugation of canine semen at 600 g for 10 min caused a significant decrease in mitochondrial membrane potential and increased lipid peroxidation [75]. A significant reduction in the natural hydrogen peroxide and hydroxyl radical concentration after removal of seminal plasma before freezing will, therefore, most probably increase post-thaw membrane lipid peroxidation unless replaced by useful substituents in the diluents. However, in ejaculates of bad quality or from old dogs, centrifugation may be beneficial, and in some protocols, pre-freeze centrifugation is routinely used to obtain a defined sperm concentration [5]. The real effect of seminal plasma removal can only be evaluated when the composition is known, as well as the composition of the ejaculate in terms of subpopulations [18].

In humans, seminal plasma membrane vesicles (MV) have been shown to fuse with the membrane of human spermatozoa, which resulted in decreased membrane fluidity in this species. In dogs, membrane vesicles of different sizes and spherical shapes were found to exert activities of variable enzymes, such as ectonucleotidases, adenosine deaminase, 5’-nucleotidase, ADPase, ATPase, contributing to ATP production and energy supply, as well as dipeptilpeptidase IV, alkaline phosphatase, total acid phosphatase and prostatic acid phosphatase activity [76]. In dogs, the interaction of MVs with the spermatozoa has not yet been proven, and the addition of different concentrations of purified MVs to canine semen before freezing did not improve post-thaw semen quality, except for a short-term improvement in distance and velocity parameters [52].

## 4. Can We Predict Sperm Freezability in Dogs?

In Table 1, an overview of possible markers of freezability is given, including principles and parameters under investigation and possible future candidates.

### 4.1. Kinematic and Morphometric Parameters—Cluster Analyses

Meanwhile, in dogs, many investigations have been performed, highlighting, among others, the importance of objective measurement of kinematic parameters [17]. Even though former studies did not provide satisfying results [13], we previously showed that kinematic parameters objectively measured by use of CASA, such as progressive motility (P), velocity curvilinear (VCL), mean coefficient (STR) and linear coefficient (LIN), are useful for the prediction of post-thaw sperm quality. Sperm samples with P < 83.1%, VCL < 161.3 µm/s, STR < 0.83% and LIN < 0.48% will have a probability of 85.5% that the post-thaw sperm quality will be low [17]. In this study, bad post-thaw quality despite good raw-semen quality (bad freezer) was defined as: progressive motility (<50%), percentage of morphological aberrations (>40%) and/or membrane integrity (<50%). However, even though the prediction is not 100%, the use of CASA can be considered a helpful step when other analyses are not possible. The restricted reliability of measurement of kinematic parameters is, among others, caused by spermatozoa subgroups. One group evaluated kinematic data obtained by CASA by use of clustering and discriminant analysis for differentiation of subgroups with different kinematic characteristics; in this study [77], 11 subpopulations were found: 4 with high velocity, 2 with medium and 5 with low velocity. The number of subgroups changed after freeze thawing. The authors state that the evaluation of subgroups is essential to demask sperms with very bad resilience toward freeze thawing that are not recognized when mean values of kinematic parameters are evaluated. Another group differentiated four subpopulations with different kinematic characteristics that were maintained after thawing [18]. Despite these differences, measurement of P, VCL, STR and LIN in raw semen combined with cluster analysis might improve the estimation of post-thaw quality.

In another study, the usefulness of computer-assisted sperm morphometry was investigated in canine spermatozoa [19]. This study impressively showed that subgroups of spermatozoa with different morphometric characteristics have variable degrees of DNA denaturation. Unfortunately, this analysis has not yet been performed in freeze-thawed canine sperm, but it proved useful for the prediction of post-thaw quality of boar semen [79].

### 4.2. Cell Volumetry

Spermatozoa volumetry was found to be indicative of the adaptability of the cell to a changing osmotic milieu, in fresh semen as well as in frozen semen [20]. Cell volume is controlled by quinine sensitive potassium channels inside the spermatozoa membrane that are functionally dependent on cytoskeleton intactness [80]. The ability of cell volume regulation is supposed to be closely linked to the ability of regulation of membrane permeability, and a loss of cell volume control before the freeze-thawing procedure coincided with an increase in cryodamage in canine spermatozoa [20]. Cell volumetry requires an electric field multi-channel cell counting system that recognizes the changes in the electric resistance caused by cells as the voltage changes; the latter are related to cell volume, as greater cells cause greater pulses. Provided such equipment is available, the method seems useful for estimation of post-thaw sperm quality in dogs. However, keeping in mind that other cryodamages not related to volume regulation may occur, additional tests may be required.

### 4.3. Seminal Plasma Components

The seminal plasma composition is species specific, and individual changes can influence freezability [16,23,50,52,66]. Meanwhile, some proteomic studies have revealed the composition of canine seminal plasma [74,81,82,83]. However, so far, no protein could be directly related to freezability of canine spermatozoa. The cholesterol content could be a potential marker [16], but more studies are necessary to prove this hypothesis. Furthermore, many assays offered by laboratories are not sensitive enough.

Products of glucose metabolism, such as lactate, might indicate a normal metabolic cell function [64], and lactate concentrations in canine seminal plasma should be related to post-thaw semen quality in future experiments. 

### 4.4. Membrane Proteins

Structural proteins of the membrane, such as the precursor of the A-kinase anchor protein 4 (ProAKAP4), stabilizing the mid-piece membrane and influencing motility, have been investigated in fresh and freeze-thawed semen. In horses, the expression of this protein was closely related to post-thaw sperm motility [84]. In pigs, the stability of proAKAP4 after thawing proved to be a quality marker [85]. In canine frozen/thawed semen, the expression of the precursor was found to be highly variable and dependent on incubation time and straw size; a correlation between proAKAP4 levels and motility or other velocity parameters could not be demonstrated [78,86]. However, it was suggested that the degree of proAKAP4 expression is related to the recovery and maintenance of velocity [78]. The latter might be interesting for the prediction of the fertilizing capability of freeze-thawed samples; however, this hypothesis has to be proven using further analyses. For the measurement of proAKAP4 in canine semen, a commercial ELISA is available.

Heat shock proteins (HSP) are important, among others, for cell protection and repair, and sperm cell maturation. These chaperons are expressed in testicular tissue and in spermatozoa of different species, and their function is not yet fully understood. In boars, the protein level of HSP90AA1 in the semen was found to be indicative of freezability [87]. In dogs, HSP60 was detectable in the mid-piece of spermatozoa, HSP70 in the neck and HSP90 in the spermatozoa tail [88]. The induction of capacitation and acrosome reaction did not change HSP70 expression in canine spermatozoa; however, the induction of acrosome reaction changed the immunosignal in boar and stallion spermatozoa [88]. The relation between HSP expression in canine spermatozoa and resilience to cryodamage remains elusive. Assessments of both HSP protein content and spermatozoa expression are of interest.

### 4.5. Response to Ionophore Treatment

Ca^2+^ ionophore is a substance enabling Ca^2+^ ions to pass the spermatozoa membrane by forming stable complexes with the divalent cations. The addition of Ca^2+^ ionophore to semen samples induces acrosome reaction, thereby enabling assessment of the cells’ functional competence. In one study [22], the percentage of acrosome reactions induced with Ca^2+^ ionophore in raw canine semen samples correlated with the percentage of acrosome reactions assessed in freeze-thawed samples; furthermore, cell damage assessed with fluorescein-conjugated peanut agglutinin (PNA-Fitc) and ethidium homodimer (EthD-1) correlated negatively with the percentage of motile cells after thawing. In another study, the increase in acrosome reactions after treatment of bull spermatozoa with Ca^2+^ ionophore was positively related to an increase in the 90-day non-return rate of inseminated cows [89], emphasizing the importance of this functional test for the prediction of the fertilizing potential of semen. However, when freeze-thawed semen from bulls was incubated with a Ca^2+^ ionophore, the correlation with fertility was low [90]. Furthermore, during an early study in humans, ionophore treatment revealed high intra- and inter-assay coefficients of variation and a high degree of intra- and inter-subject variability [91]. This has to be considered in addition to the fact that the effect of Ca^2+^ ionophore treatment is dependent on concentration and duration of incubation. This has not been sufficiently investigated in canines, and more studies are needed.

## 5. Conclusions and Outlook

The at-present most useful combination for a quick prediction of freezability appears to be a combination of kinematic parameters and DNA integrity. If possible, an examination by CASA should be combined with cluster analysis of kinematic subgroups, for better recognition of the causes of bad freezability, and with JC-1 staining and a simple DNA integrity assay, such as the COMET assay or staining with toluidine blue. Cell volumetry is useful but requires special equipment. Seminal plasma parameters require more investigation before application in practice; proteomic analyses of seminal plasma of bad and good freezers will probably reveal further markers. Further markers of freezability that proved to be useful in other species, especially those indicative of energy metabolism, mitochondria activity and DNA intactness, deserve better attention. In the canine species, the genetic aspect may be underestimated. Bad semen quality has been associated with inbreeding and the bottleneck effect in wild species. In bulls, genetic diversity was positively correlated with post-thaw motility and viability [92]. In this species, some microsatellite markers and single nucleotide polymorphisms (SNPs) were found to be related to post-thaw motility [30]. No study is available investigating possible genetic effects on post-thaw canine sperm quality in the relevant literature, rendering breed-specific studies using microsatellites and SNPs highly interesting. However, the practical and economical applications of freezability markers must be considered. 

This review is a snapshot of the present situation. A better understanding of spermatozoa function, especially energy metabolism, might reveal new markers, and the improvement of assays will probably simplify the combined assessment of markers in future.

## Figures and Tables

**Figure 1 animals-12-00733-f001:**
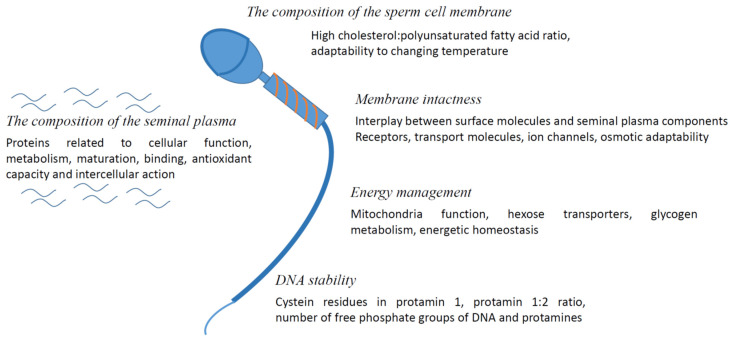
What makes sperm freezable?

**Table 1 animals-12-00733-t001:** Predictors of freezability of canine semen.

Principle	Parameter	Assay/Method	Authors
Kinematic data and cluster analyses	Kinematic data	CASA	[77] [17]
DNA integrity assays	DNA damage	Comet assay, SCSA, Toluidine blue stain	[68] [66] [72]
Cell volumetry	Adaptability to changing osmotic milieu	Electric field multi-channel cell counting system	[20]
Mitochondria assays	Inner mitochondrial membrane potential, *Mitochondria function* *ATP production*	Flow cytometry, ICC (JC-1 probe) *Commercial assays*	[63]
Induction of acrosome reaction by Ca^2+^ ionophore	Acrosome reaction	Addition of Ca^2+^ ionophore	[22]
*Membrane components*	*proAKAP4*	*Commercial ELISA*	[78]
	*Aquaporin 1, 8*	*ICC*	[60,61]
	*Hexose transporters*	*ICC*	[55]
*Seminal plasma components*	*Cholesterol*	*Chemiluminescence*	[16]
	*Lactate*		[64]
*Transcriptome analysis*	*Identification of genes related to good/bad freezability*		
*Microsatellite markers, SNPs*	*Post-thaw motility, viability, fertility*		

Principles and parameters under investigation/future candidates are in italics. CASA = computer assisted semen analysis, HSP = heat shock protein, ELISA = enzyme-linked immunosorbent assay, ICC = immunocytochemistry, proAKAP4 = precursor of the A-kinase anchor protein 4, SCSA = sperm chromatin structure assay, SNP = single nucleotide polymorphism.

## Data Availability

No new data were created or analyzed in this study. Data sharing is not applicable to this article.

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
