# Peer review of "Canine Spermatozoa—Predictability of Cryotolerance"

_animals, 2022, doi:10.3390/ani12060733_

Round 1

Reviewer 1 Report

Canine semen freezing is an inseparable part of canine reproduction. Therefore high quality of frozen semen is desired. However, the quality of semen after defrosting is always much lower than of the fresh semen. Markers of freezability could greatly help to estimate this change.

The manuscript is very well written and concerns important issues of veterinary medicine reproduction. It summarizes important predictors of canine semen freezability, however some minor issues should be explained:

  1. In the text aquaporin 1 is mentioned. What about other aquaporins? To my knowledge, other aquaporins such as for example aquaporin 7 or aquaporin 9 have also been detected in canine semen and may play an important role.
  2. What about the influence of extender composition and freezing protocol on canine semen freezability? Shouldn’t this also be discussed?

Clinical studies in this field are urgently needed.

Author Response

Canine semen freezing is an inseparable part of canine reproduction. Therefore high quality of frozen semen is desired. However, the quality of semen after defrosting is always much lower than of the fresh semen. Markers of freezability could greatly help to estimate this change.

The manuscript is very well written and concerns important issues of veterinary medicine reproduction. It summarizes important predictors of canine semen freezability, however some minor issues should be explained:

  1. In the text aquaporin 1 is mentioned. What about other aquaporins? To my knowledge, other aquaporins such as for example aquaporin 7 or aquaporin 9 have also been detected in canine semen and may play an important role.

Authors: we found additional information on AQP-8 but not on -7 or -9 in canine spermatozoa. We inserted the respective information in the overworked manuscript:

“A defective membrane will among others affect the function of principle water channel molecules. Aquaporins (AQP) exert a protective effect during the freeze-thaw procedure, not only by active osmoregulation but also by regulting the transport of small solutes like cryoprotective agents (CPA) [57]. They are widely distributed in testicular and epididymal tissue [58], and were detected in spermatozoa membranes of human, mouse, rat, boar, stallion, seabream, goose and bull (for review: [57]); however, in a species-specific and sometimes even individual manner.

The impact of water channel molecules on semen freezability has been investigated in other species [59] and expression of different AQP was related to the cells’ ability to survive the cryopreservation procedure which seems to be species-specific [57]. Recently, AQP-1 has been detected in the membrane of canine spermatozoa, and was localized in the head, midpiece and tail [60]. In another study, AQP-8 was detected in canine spermatozoa by means of immunoblotting [61], unfortunately, assessment of the site of expression was thereby not possible. The relative amount of AQP-8 was positively correlated with the percentage of proximal cytoplasmic droplets [61]. Being an orthodox AQP, AQP-8 is supposed to mainly contribute to cell volume regulation; however, it is not only permeable to water and mitochondrial AQP-8 is in addition able to transport hydrogen peroxide. A role for AQP-8 in diffusion of ROS after cryopreservation has been suggested previously [57]. The function of AQP-8 in canine spermatozoa membranes and its usefulness as a marker of freezability remain to be investigated.

  1. What about the influence of extender composition and freezing protocol on canine semen freezability? Shouldn’t this also be discussed?

Authors: We briefly mention this in the introduction, however, referring to the title of this review, intense discussion of extender composition and freezing protocols would burst the frame of this review. However, we inserted more information and literature:

The freezability of a canine spermatozoa is determined by the complex interplay of sperm membrane and seminal plasma, diluent and cooling-freezing-thawing protocols [1–6], and is influenced by further parameters like age [7]. The sensitivity to cryoinjuries is species-specific and recent reviews comprehensively summarized the requirements for successful cryopreservation of canine spermatozoa [1,8]. Canine spermatozoa membranes are less sensitive to cold-injury than for example boar spermatozoa, due to a relatively high cholesterol: phospholipid ratio [9] and the polyunsaturated fatty acid (PUFA) content of the membrane. This is among others indicated by the fact that DNA integrity is not affected by freezing-thawing [10], whereas in stallions, it is [11]. Unfortunately the exact composition of the sperm membrane PUFA has not yet been unraveled in dogs, as in other species [12]. However, semen quality after thawing in addition is individual-dependant.

Reviewer 2 Report

Dear Authors

First of all, I convey my sincere greetings to all of you for writing this review in the area of fertility and semen markers for canines. Although you attempted to address one of the major issues in the area of canine semen research, I am sorry to find the novelty in this area in the present form of this manuscript. Therefore, I request you to go through my suggestions and try to append them to the manuscript after an extensive revision.

  1. Title: I found this title is not suitable as per the content of the manuscript and needs changes so as to be more appropriate for this review.
  2. Abstract: Is well written, however, the authors are failed to mention the prognostic markers of canine semen freezing.
  3. Introduction: Well designed and written. The only issue which I observed is- the authors failed to build the hypothesis behind this review means the background of this review. How this review will enhance the understanding of the existing subject? What new markers the authors are suggesting for using them as freezing markers in canines? The most important part of it is the background information regarding the framing of this review and this needs to be added in the introduction.
  4. The major issue which I came across is the generalized description of the events which occur during freezing-thawing in spermatozoa. These things are well known and multiple reviews are currently available. The authors described these events in detail like- cryopreservation and membrane damage, semen freezability parameters, the membrane composition and intactness, energy management and DNA stability, and seminal plasma composition. I want to know whether these parameters are different from other animals or they are similar in canines and other species of animals. I would rather suggest the authors mention those things which are relevant to the canine semen only which I found missing in the current form of the manuscript.
  5. The best part of this review for the readers is the predictor’s part mentioned in Table 1. This table serves as the key for determining the different aspects of semen-freezable parameters required for canine semen. This table should be discussed in more detail and authors should derive solid conclusions out of these studies to include in this present review.
  6. I can suggest that this review may be presented in the form of a mini-review as limited information is available in the area of canine semen freezing. I would rather suggest the authors quote the proteomics and transcriptomic markers in the review of any other animals reported as they are not available for canine semen.
  7. Further, it is a suggestion for the authors to include a physiological aspect of semen and sperm physiology in the context of canine semen and a comparative analysis with other animals (if any). Thereby, the readers can get new information out of this review in terms of canine semen freezing and the biomarkers.

Wishing you good luck….

Author Response

First of all, I convey my sincere greetings to all of you for writing this review in the area of fertility and semen markers for canines. Although you attempted to address one of the major issues in the area of canine semen research, I am sorry to find the novelty in this area in the present form of this manuscript. Therefore, I request you to go through my suggestions and try to append them to the manuscript after an extensive revision.

  1. Title: I found this title is not suitable as per the content of the manuscript and needs changes so as to be more appropriate for this review.

Authors: we regret that this reviewer does not like the title, however, if not acceptable at all, we suggest the following:

“Canine spermatozoa – predictability of cryotolerance”

  1. Abstract: Is well written, however, the authors are failed to mention the prognostic markers of canine semen freezing.

Authors: the abstract was rewritten and the most useful combination of assays inserted:

Markers of freezability allow the selection of ejaculates of good freezability. So far most investigations were done in boars, bulls, rams and horses, with high economic interests triggering the efforts. The progress in dogs is comparably slow. Critical evaluation of methods requires consideration of practicability, with most labs not even possessing CASA; furthermore small canine ejaculates mostly do not allow the use of large semen volumes. In dogs, modern markers of freezability no longer assess single membrane constituents or seminal plasma components, but comprise tests of cell functionality and adaptability, energy metabolism, cluster analyses of kinetic and morphometric parameters as well as DNA intactness. Identification of the most efficient combination of tests seems useful. At present, examination by CASA combined with cluster analysis of kinetic subgroups, JC-1 staining, and COMET assay or staining with toluidine blue seem most appropriate; however, cell volumetry and other functional tests deserve better attention. A better understanding of spermatozoa energy metabolism might reveal new markers. This review focuses on the requirements and markers of freezability of canine semen, highlighting potential future candidates

  1. Introduction: Well designed and written. The only issue which I observed is- the authors failed to build the hypothesis behind this review means the background of this review. How this review will enhance the understanding of the existing subject? What new markers the authors are suggesting for using them as freezing markers in canines? The most important part of it is the background information regarding the framing of this review and this needs to be added in the introduction.

Authors: we agree that this can be improved and overworked the chapter. Now the first paragraph already emphasizes the background:

The freezability of a canine spermatozoa is determined by the complex interplay of sperm membrane and seminal plasma, diluent and cooling-freezing-thawing protocols [1–6], and is influenced by further parameters like age [7]. The sensitivity to cryoinjuries is species-specific and recent reviews comprehensively summarized the requirements for successful cryopreservation of canine spermatozoa [1,8]. Canine spermatozoa membranes are less sensitive to cold-injury than for example boar spermatozoa, due to a relatively high cholesterol: phospholipid ratio [9] and the polyunsaturated fatty acid (PUFA) content of the membrane. This is among others indicated by the fact that DNA integrity is not affected by freezing-thawing [10], whereas in stallions, it is [11]. Unfortunately the exact composition of the sperm membrane PUFA has not yet been unraveled in dogs, as in other species [12]. However, semen quality after thawing in addition is individual-dependant. In some dogs, despite good quality of raw semen, post-thaw semen quality is bad [13–16]. This rectifies the intense search for prognostic markers of freezability. To date, mainly kinematic and morphometric parameters including cluster analyses [17–19], functional assays like volumetry [20], the hyposomotic swelling test (HOST, [21]) or the reaction to (Ca2+) ionophore treatment [22], and seminal plasma components [16,23] have been investigated; however the usefulness for prediction of post-thaw semen quality has not been reviewed in a comparative manner so far. In dogs, best results were achieved with functional tests like kinematic assays combined with cluster analyses, cell volumetry and the reaction to (Ca2+) ionophore, whereas the HOST was not useful to predict post-thaw semen quality [21]. This is comparable to other species [24] and highlights the need for complex assays, investigating more than adaptability to changing osmolarities.

In other species, abundance of markers for freezability have meanwhile been detected that have not yet been investigated in canines. Especially in boars, investigations are intense (for review: [25]). Only recently, some spermatozoa components of metabolic pathways related to energy metabolism were identified as probable suitable markers [26,27]. Another study using transcriptome analysis revealed an upregulation of several genes, and finally a higher concentration of 3 proteins related to inflammation and apoptosis in bad freezer ejaculates [28]. Comparable studies in dogs are lacking. In bulls, proteomic approaches revealed higher concentration of Arylsufatase A in semen of good freezers, which is indicative for the cell’s energy metabolism. Further markers of sperm functionality were assessed in high density spermatozoa, indicative for normal glycolysis, zona binding and motility [29]. In this species, meanwhile genome wide association studies investigate genetic causes of poor sperm quality, and microsatellites markers (SNPs) like BM1500 and UMN2008 were found to be related to freezability, especially post-thaw motility [30]. In rams, a strong correlation between seminal plasma composition and freezing resilience of semen has been found [31]. In stallions, among others, caspase 3 activity and the lipoperoxidative status of semen were found to be suitable for the prediction of freezability [32]; as well as the content of cysteine-rich secretory proteins (CRISP-3), participating in sperm maturation [33]. In men, spermatozoa out of bad freezer ejaculates contained less enolase 1 and glucose-6-phosphate isomerase (GPI) than cells from good freezer ejaculates. Both parameters are among others indicators of glycolysis and energy production [34]. Furthermore, viscosity of fresh semen was negatively related to post-thaw motility and acrosome integrity, and low citric acid concentration was negatively related to post-thaw acrosome integrity [35].

Markers of freezability are thus mainly indicators of spermatozoa functionality and energy metabolism, or seminal plasma components. The present review discusses the usefulness and reliability of markers of freezability of canine spermatozoa, and highlights potential factors of future interest.

  1. The major issue which I came across is the generalized description of the events which occur during freezing-thawing in spermatozoa. These things are well known and multiple reviews are currently available. The authors described these events in detail like- cryopreservation and membrane damage, semen freezability parameters, the membrane composition and intactness, energy management and DNA stability, and seminal plasma composition. I want to know whether these parameters are different from other animals or they are similar in canines and other species of animals. I would rather suggest the authors mention those things which are relevant to the canine semen only which I found missing in the current form of the manuscript.

Authors: we agree and inserted some sentences in this chapter, emphasizing the findings in canine spermatozoa in comparison to other species (marked in blue in the overworked manuscript)

  1. The best part of this review for the readers is the predictor’s part mentioned in Table 1. This table serves as the key for determining the different aspects of semen-freezable parameters required for canine semen. This table should be discussed in more detail and authors should derive solid conclusions out of these studies to include in this present review.

Authors: we in fact very often refer to the canine species and highlight species-specific findings; however, we tried to overwork this chapter accordingly

  1. I can suggest that this review may be presented in the form of a mini-review as limited information is available in the area of canine semen freezing. I would rather suggest the authors quote the proteomics and transcriptomic markers in the review of any other animals reported as they are not available for canine semen.

Authors: we do not agree since the aim of this review is to discuss the usefulness of markers of freezability in canines in comparison to other species. This is far too much for a mini review.

  1. Further, it is a suggestion for the authors to include a physiological aspect of semen and sperm physiology in the context of canine semen and a comparative analysis with other animals (if any). Thereby, the readers can get new information out of this review in terms of canine semen freezing and the biomarkers.

Authors: we agee that this would be interesting but it is our impression that this additional chapter would increase the scientific content too much and burst the frame of this review.

Reviewer 3 Report

Although canine specific, this review by Schäfer-Somi et al. would be useful to any andrologist interested in cryopreservation of semen in any animal system. The article is comprehensive, but it does not explain many features in detail or even all of the available techniques for the beginners in seminal cryobiology. For example, section 4.5 could contain more details as well as some aspects of the mechanism. It is currently only supported by a single reference. I am also unable to see a note in the article on a live-dead staining technique that is widely used to assess membrane damage, namely Sybr-14/propidium iodide (https://doi.org/10.1095/biolreprod53.2.276).

In general, some sentences in the manuscript are not well structured. A non-author of this manuscript should read and make the manuscript convey information better.

Sentence in Line 99-100: Generally high cholesterol and low PUFA tend to make the membrane more rigid especially at low temperatures and consequently result in leaky membrane. (Reynolds, A.M., Lee, R.E. & Costanzo, J.P. J Comp Physiol B 184, 371–383 (2014).). Hence the current understanding of the nature of cryoprotection by cholesterol is, that it helps maintain membrane intactness at non-cryogenic temperatures, primarily during cold shock which is usually at above freezing temperatures (Huebinger J (2018)  PLoS ONE 13(10): e0205520). I think this should be noted in the manuscript.

164 - It is a Raman spectrometer. On another note, Raman spectroscopy requires isolated DNA, unless the authors show otherwise, if not this should be stated. The isolation process of the DNA by itself will generate artifacts and it is often cumbersome to parse the artifacts between the treatment and controls.

Sentence 164-165 - Please rephrase the sentence starting with 'Andrological'.

Line 170: Requires a comma -  ... seminal plasma of good freezers, higher concentrations...

Line 240 - ...post-thaw quality of boar semen.

Line 258: Replace ‘unscrambled’ with ‘revealed’. Seminal plasma composition is not scrambled firstly to be unscrambled.

Line 278 - Authors need to explain what a "testis chaperon" is. HSPs are often described as molecular chaperons for such molecules as enzymes, especially under stressful conditions.

Author Response

Although canine specific, this review by Schäfer-Somi et al. would be useful to any andrologist interested in cryopreservation of semen in any animal system. The article is comprehensive, but it does not explain many features in detail or even all of the available techniques for the beginners in seminal cryobiology. For example, section 4.5 could contain more details as well as some aspects of the mechanism. It is currently only supported by a single reference.

Authors: we agree and inserted a respective paragraph with more information:

In another study, the increase in acrosome reactions after treatment of bull spermatozoa with Ca(2+)ionophore was positively related to an increase in the 90-day non-return-rate of inseminated cows [90]; emphasizing the importance of this functional test for the prediction of the fertilizing potential of semen. However, when frozen-thawed semen from bulls was incubated with a Ca(2+)ionophore, correlation with fertiliy was low [91]. Furthermore, during an early study in humans, ionophore treatment revealed high intra- and inter-assay coefficients of variation and a high degree of intra- and inter-subject variability [92]. This has to be considered and in addion that the effect of Ca(2+) ionophore treatment is dependant on concentration and duration of incubation. Both has not been sufficiently investigated in canines and more studies are needed.

 I am also unable to see a note in the article on a live-dead staining technique that is widely used to assess membrane damage, namely Sybr-14/propidium iodide (https://doi.org/10.1095/biolreprod53.2.276).

Authors: we included this information in chapter 3.2. in the overworked manuscript:

Assessment of membrane integrity is therefore a useful part of pre-freeze semen evaluation. Fluorescent dys like SYBR-14/PI and CFDA enable visualisation of damaged membranes [53], whereas evaluation of specifically acrosome damages with fluoresceinated lectin peanut agglutinin was found to be extremely valuable to assess sperm quality before freezing and after thawing in dogs and other species [20,54].    

In general, some sentences in the manuscript are not well structured. A non-author of this manuscript should read and make the manuscript convey information better.

Sentence in Line 99-100: Generally high cholesterol and low PUFA tend to make the membrane more rigid especially at low temperatures and consequently result in leaky membrane. (Reynolds, A.M., Lee, R.E. & Costanzo, J.P. J Comp Physiol B 184, 371–383 (2014).). Hence the current understanding of the nature of cryoprotection by cholesterol is, that it helps maintain membrane intactness at non-cryogenic temperatures, primarily during cold shock which is usually at above freezing temperatures (Huebinger J (2018)  PLoS ONE 13(10): e0205520). I think this should be noted in the manuscript.

Authors: we inserted this valuable information

164 - It is a Raman spectrometer. On another note, Raman spectroscopy requires isolated DNA, unless the authors show otherwise, if not this should be stated. The isolation process of the DNA by itself will generate artifacts and it is often cumbersome to parse the artifacts between the treatment and controls.

Sentence 164-165 - Please rephrase the sentence starting with 'Andrological'.

Authors: we considered the reviewer’s concerns: " A rather sophisticated approach, however, promising is the Raman spectrometer; this technique relies on the interaction of light photons with molecules, resulting in different light-scattering patterns that can be immediately evaluated using semen or seminal fluid. Some applications are the differentiation of abnormal and normal seminal plasma, and DNA damage [73].  „

Line 170: Requires a comma -  ... seminal plasma of good freezers, higher concentrations...

Authors: this was changed accordingly

Line 240 - ...post-thaw quality of boar semen.

Authors: this was changed accordingly

Line 258: Replace ‘unscrambled’ with ‘revealed’. Seminal plasma composition is not scrambled firstly to be unscrambled.

Authors: this was changed accordingly

Line 278 - Authors need to explain what a "testis chaperon" is. HSPs are often described as molecular chaperons for such molecules as enzymes, especially under stressful conditions

Authors: we changed the sentence: These chaperons are expressed in testicular tissue and in spermatozoa of different species, and their function is not yet fully understood.

Round 2

Reviewer 2 Report

Dear Authors

I am happy to see the changes and incorporations in the manuscript in the line of reviewer's suggestions. I am happy with the content of the manuscript as well. Most of the suggestions have been addressed. I also concur with the previous title of the review and you can keep it.

with best wishes